# STOP leg clots—Swedish multicentre trial of outpatient prevention of leg clots: study protocol for a randomised controlled trial on the efficacy of intermittent pneumatic compression on venous thromboembolism in lower leg immobilised patients

Simon Svedman [1,2] Björn Alkner,[3] Hans E Berg,[1,4] Erica Domeij-Arverud,[5] Kenneth Jonsson,[6] Katarina Nilsson Helander,[7] Paul W Ackermann[1,2]

**Correspondence to**
Paul W Ackermann;
Paul.ackermann@ki.se

## ABSTRACT

**Introduction** Leg immobilisation in a cast or an orthosis after lower limb injuries is associated with a high risk of complications of venous thromboembolism (VTE) and hampered healing. Current pharmacoprophylaxes of VTE are inefficient and associated with adverse events. Intermittent pneumatic compression (IPC) could represent a novel, efficient and safe VTE-prophylactic alternative that may enhance injury healing. The aim of STOP leg clots is to assess the efficacy of adjuvant IPC-therapy on reduction of VTE incidence and improvement of healing in lower leg immobilised outpatients.

**Methods and analysis** STOP leg clots is a multicentre randomised controlled superiority trial. Eligible patients (700 patients/arm) with either an acute ankle fracture or Achilles tendon rupture will be randomised to either addition of IPC during lower-leg immobilisation or to treatment-as-usual. The primary outcome will be the total VTE incidence, that is, symptomatic and asymptomatic deep venous thrombosis (DVT) or symptomatic pulmonary embolism (PE), during the leg immobilisation period, approximately 6–8 weeks. DVT incidence will be assessed by screening whole leg compression duplex ultrasound at removal of leg immobilisation and/or clinically diagnosed within the time of immobilisation. Symptomatic PE will be verified by CT.

Secondary outcomes will include patient-reported outcome using validated questionnaires, healing evaluated by measurements of tendon callus production and changes in VTE-prophylactic mechanisms assessed by blood flow and fibrinolysis. Data analyses will be blinded and based on the intention-to-treat.

**Ethics and dissemination** Ethical approval was obtained by the ethical review board in Stockholm, Sweden, Dnr 2016/1573-31. The study will be conducted in accordance with the Helsinki declaration. The results of the study will be disseminated in peer-reviewed international journals.

**Trial registration** NCT03259204.

**Time schedule** 1 September 2018 to 31 December 2022

## Strengths and limitations of this study

► This is the first multicentre randomised controlled trial to investigate the efficiency of adjuvant intermittent pneumatic compression on venous thromboembolism (VTE) for the duration of ankle joint immobilisation in an outpatient setting.

► The multicentre design improves the external validity of the study and will allow rapid dissemination and potential implementation of study results.

► Patients cannot be blinded to allocation.

► Clinical sonographers are responsible to assess the primary outcome and will be blinded to patient allocation.

► The duplex ultrasound screening will detect subclinical VTE which will allow high study power. However, the clinical relevance of non-symptomatic events is still not fully clear.

## INTRODUCTION

Venous thromboembolism (VTE), comprising deep venous thrombosis (DVT) and pulmonary embolism (PE), is a critical medical condition and potentially deadly. An estimated 370 000 people die of VTE in Europe each year, and more than 700 000 Europeans are afflicted.[1] Patients who survive often suffer chronic ailments with significantly lower quality of life after an event.[2–7] Leg immobilisation in a cast or orthosis is a routine stage in the treatment of patients with lower limb injury, which however causes high rates of VTE due to venous stasis.[8] Estimated DVT incidence may range from 21% up to 50% after ankle fractures (AF) and Achilles tendon ruptures (ATR), respectively.[9 10]

The most widely used method to prevent VTE during lower leg immobilisation is by the use of pharmacological prophylaxis, such as low molecular weight heparin (LMWH). However, several studies have demonstrated only a marginal or no effect of LMWH in lower leg immobilised patients and LMWH is therefore not cost efficient for these patients.[7 11–15] In addition to their low efficacy, LMWH increases the risk of bleeding complications[13 14] and may impair tissue healing.[16] For these reasons, the National Health Service in UK, and the corresponding authority in Sweden, have stressed the need for safer and more efficient VTE prophylaxis.[17–19]

Mechanical thromboprophylaxis with intermittent pneumatic compression (IPC) reduces VTE is a cost-effective intervention in hospitalised patients, and is recommended by guidelines for surgical in-hospital patients.[17 20] Whether long-term IPC treatment applied during leg immobilisation in an outpatient setting can reduce the incidence of DVT is unknown. We recently published a study presenting preliminary evidence that leg immobilised patients with ATR treated with adjuvant IPC at 2 weeks had a reduced DVT rate (21% vs 37%) compared with controls using Doppler ultrasound screening. However, in the previous study, the IPC intervention ended at 2 weeks and at 6 weeks, the DVT rate was 50% in both groups.[9]

Thus, the primary aim of this prospective, multicentre, randomised controlled trial is to evaluate the efficacy of prolonged adjuvant IPC therapy in leg-immobilised patients on the total incidence of VTE up to removal of leg immobilisation.

Lower leg immobilisation, in addition to increased risk of VTE, may also lead to impaired healing after initial injury as well as long-lasting muscle atrophy.[21] As recent studies suggest that the increased neuro-vascular flow induced by IPC could trigger the healing process,[22] we also aim to assess whether adjuvant IPC enhances healing, reduces muscle atrophy and improves the long-term patient-reported outcomes in lower limb immobilised patients.

## METHODS AND ANALYSIS

This is a report of the trial protocol dated 4 September 2017, supported by the Swedish research council. Any important protocol modifications will be communicated through ClinicalTrials.gov trial number NCT03259204, the Ethical Review Board and/or the BMJ Open journal. The protocol was developed in accordance with Standard Protocol Items: Recommendations for Interventional Trials and Template for Intervention Description and Replication statements.[23]

## Study design and setting

'STOP leg clots' is a multicentre, prospective, superiority, randomised, controlled, parallel group trial with blinded assessment of the outcome.

## Recruitment organisation

In general, patients will be informed about the study at their visit to the emergency department. As this is a multicentre trial, different methods will be suitable and used by different participating study-sites in order to include patients. Patients who are not subjected to further treatment in the hospital will either be included and randomised immediately after given both written and oral information about the study, or if this is not possible, receive information about the study and scheduled for an outpatient study appointment within the following days. Patients who are in need of surgery and operated on within 10 days can be included and randomised, after receiving written and oral information. All participating patients will sign an informed consent form, which will be collected by the local study personnel and delivered to the central study organisation for archiving and the end of the trial. The steering committee will aid recruiting centres in setting up the local infra-structure needed for patient recruitment.

## Study organisation

11 hospitals from seven different Swedish regions and two hospitals from two other European countries have so far accepted to participate in the study.

## Study participants

1400 patients will be included; where approximately 1000 AF and 400 ATR.
► Inclusion criteria: age of 18–75 years, unilateral isolated AF or ATR and able to be included in the study within 10 days of injury.
► Exclusion criteria: unable to give informed consent, unable to follow instructions and unable to receive orthosis treatment, known kidney failure, heart failure with pitting oedema, malignancy, haemophilia, thrombophilia, current DVT or PE, pregnancy, or has follow-up planned at a hospital outside of the study organisation.

## Objective

The primary objective is to investigate if adjuvant IPC treatment during the duration of lower leg immobilisation treatment will reduce the incidence of VTE in an outpatient setting.

## Randomisation

Patients will prior to randomisation receive both written and oral information and sign an informed consent form. Patients will be randomly assigned to either standard treatment or standard treatment with adjuvant IPC in an orthosis with a 1:1 allocation. Patient randomisation may occur up to 10 days after the injury. The number of days from injury to start of randomised treatment as well as potential immobilisation will be noted. The randomisation is stratified by study site, patient age over/under 40 years and by type of injury (AF or ATR). Randomisation occurs in blocks of 4. Randomisation lists are uploaded to the REDCap online eCRF system[24] hosted at Karolinska

Institute, only accessible to the REDCap administrator of the trial. Randomisation is only performed by instructed study personnel. All activities within REDCap are tracked, and the REDCap log is non-modifiable.

## Intervention

The intervention, *addition of IPC to routine care*, will be administered in the emergency department, the outpatient clinic or the hospital ward. Patients randomised to the intervention will during the whole duration of lower limb immobilisation receive bilateral calf-IPC (Venaflow Elite, DJO LLC, Vista, CA, USA), on the injured side applied under an orthosis (Aircast Airselect Elite Walker, DJO LLC, Vista, CA, USA). Calf-IPC has previously been successfully used under an orthosis.[9] Patients wearing an orthosis are instructed to open the orthosis 2–3 times per day while sedentary to reduce the humidity in the orthosis and prevent blistering. When IPC is used, the calf-cuffs will alternately inflate every 30 s. The patients will be instructed to apply the IPC-therapy as much as possible during the time they are sedentary, both day and night, with a goal of 10 hours/24 hours. Patients should, however, not keep themselves from being active in order to comply with IPC-treatment. Compliance with the treatment will be registered by both the patient and by the device.[9] IPC will be discontinued if the patient declines to continue IPC or had adverse effects of the IPC that warrants removal. Patients who use the IPC for 30 hours/week or more will be deemed compliant to the treatment. IPC treatment will be ended when the leg immobilisation is removed.

## Comparator

Patients randomised to standard care will receive lower limb immobilisation in an orthosis or a below-knee plaster cast according to local routines. A previous study did not show any difference in the incidence of VTE between plaster cast immobilisation and orthosis-treatment.[25] As standard care varies according to local routines and type of injury regarding duration of immobilisation as well as weight-bearing these factors will be recorded and analysed for.

Intake of pharmacological anti-thrombotic drugs before the injury or administered during the immobilisation period will be registered for category, dosage and duration. There is no general recommendation for prescribing VTE-prophylactic drugs in the trial for patients without any known risk factors for thrombosis.

## Study outcomes

### Primary outcome measure

The primary outcome is the incidence of VTE during the lower leg immobilised period. VTE will be assessed clinically if a patient develops symptoms of DVT or PE, which is verified by compression Duplex ultrasound or CT, respectively. Moreover, DVT will be screened for by whole leg Compression Duplex ultrasound (CDU) of the injured leg at termination of leg immobilisation by an experienced sonographer, which is 6–8 weeks after injury, depending on the type of injury and treatment used. The patient will furthermore be asked to answer if a symptomatic VTE has occurred after the initial 6–8 weeks follow-up in the survey at 12 months.

### Secondary outcome measure

Secondary outcomes planned for the study are listed below. Many of the secondary outcome measures need a much smaller number of patients included and will therefore only be performed on a limited number of sites to decrease the risk of methodological errors.

Secondary outcome measures include the following:

► Patient-Reported Outcome Measures will be reported by all participating patients at 6 and 12 months after injury. The questionnaires EQ-5D-5L[26] and the Foot and Ankle Outcome Score[27] will be answered by all patients. In addition, the Olerud-Molander Outcome Score[28] will be answered by the AF patients and the Achilles tendon Total Rupture Score[29] by the ATR patients. The data will be collected and managed using REDCap electronic data capture tool.[24]

► Compliance to IPC treatment. Patients randomised to IPC treatment will give weekly reports regarding IPC-device usage by daily journals. The IPC device will also record the cumulative time used.

► Postinjury weight-bearing. Weight-bearing after injury have been suggested to improve lower limb-blood flow and decrease risk of DVT.[25] Weight-bearing will be recorded via patient recorded daily journals. The patient is asked to note how much weight they load on their injured foot while walking, on a scale from 0 to 100. 0 is equal to non-weight-bearing walking with the aid of crutches and 100 is fully weight-bearing in the orthosis/plaster cast on the injured foot without crutches. A recent study has shown that this scale analysis is a reliable method of assessing weight-bearing of the patient.[30]

► Adverse events. Events such as postoperative infection, fracture dislocation, rerupture or reoperation will be recorded. Furthermore, patient mortality will be registered at 1 year after injury.

► IPC-related adverse events. Events caused by IPC treatment such as skin damage or infection, prematurely discontinuing IPC treatment or fall injuries will be recorded.

► Microdialysis of Achilles tendon healing. Healing metabolites as well as callus production will be assessed at termination of leg immobilisation, 6–8 weeks, using microdialysis in a subgroup of 50 ATR patients, followed by quantification of procollagens, to assess tendon healing. Microdialysis is a well-known process for this aim, where a small catheter is inserted in the paratenon space.[31] Samples will be collected and stored at Stockholm medical biobank.

► Heel-rise endurance test. The heel-rise endurance test is an established functional test for patients with ATR and will be performed by a subgroup of patients at

1 year after injury to measure muscle-tendon strength in the injured, compared with the uninjured leg. The MuscleLab (Ergo Test Technology, Stathelle, Norway) measurement system is used.

► VTE-preventive mechanisms. Assessment of fibrinolysis and coagulation factors as well as blood-flow quantification using ultrasound will be assessed in a subgroup of 50 patients. Overall, haemostatic potential, D-dimer, Plasminogen activator inhibitor-1, endogen thrombin potential, trombin-antitrombin-complex, plasmin antiplasmin complex and tissue plasminogen activator/PAI complex will be assessed at Karolinska Institute, Coagulation Unit. Samples will be collected from patients willing to participate at discontinuation of lower limb immobilisation, prepared and stored at Stockholm medical biobank until analysis.

► The incidence of VTE in patients using anticoagulants such as LMWH with or without adjuvant IPC will be analysed.

► Blood-flow quantification. Peak popliteal and femoral blood velocity will be calculated using custom ultrasound software and compared with baseline values, while microcirculation and tissue oxygenation will be assessed by near-infrared spectroscopy (INVOS). These explanatory continuous data will be compared between the groups.

► Cost–benefit analysis. Patient annual salary, type of occupation and the numbers of days on sick-leave will be recorded. These data will be used in calculating cost-effectiveness for the prevention of VTE.

## Blinding

Blinding will not be possible for the patient or for the study personnel responsible for patient inclusion. Sonographers will be blinded to patient allocation since the immobilisation will be removed prior to ultrasound examination. All patients are instructed not to disclose randomisation allocation to the sonographers, and the sonographers are informed not to ask about this information.

## Time schedule

Recruitment of patients began on 1 September 2018. Estimated primary completion date is 31 December 2022.

## Setup
### Follow-up investigations

All patients will be followed as usual with an outpatient visit approximately at 2 weeks after injury/surgery where compliance and adverse events will be recorded. At the time of termination of treatment (usually 6–8 weeks after injury/surgery), the orthosis will be removed and the patient will be examined by an orthopaedic surgeon or a physiotherapist, who will document compliance and adverse events. At this time, the ultrasound investigation of the lower limbs will occur, following a standardised

protocol. The patient will be treated according to local routines in the event of a venous thrombosis.

Patients will receive a journal where they will note the estimated amount of loading on their injured leg (0%–100%) and, if randomised to intervention, how many hours they used IPC treatment each day. This will be self-reported via the REDCap digital clinical report forms once a week and rechecked at the last outpatient when discontinuing the lower leg immobilisation. Patients who do not have access to an e-mail address or do not want to report their data digitally will hand in their journal at the discontinuation of immobilisation, and the data will be entered manually by the research personnel.

### Protocol for duplex ultrasound

Ultrasound will be performed throughout by trained sonographers on the injured leg only. Locations investigated include the deep veins (femoralis communis, femoralis superficialis, femoralis profunda, poplitea, tibialis posterior and fibularis) and muscular veins. A positive investigation is reported when any of the following is observed: visible thrombi in the 2D projection, a non-compressible vein, an expanded vein and lack of Doppler signal or stripe flow. All positive findings will be validated by a sonographer or radiologist in the steering-committee.

## Statistical analysis
### Sample size

The STOP leg clots is designed on a hypothesis of superiority of adjuvant IPC therapy compared with standard treatment during lower leg immobilisation in prevention of VTE. Based on the existing literature, the expected occurrence of DVT in the control group as detected by screening ultrasound is up to 25% for AF and 50% in ATR.[9 10] The estimates are dependent on the case-mix ratio as the incidence of DVT is higher in ATR than in AF. We expect the case-mix to be 30% ATR and 70% AF, resulting in an expected DVT rate of 33%. As extra precautions, we reduced the expected total DVT rate from 33% to 28%. The expected relative risk reduction of DVT is 40% based on earlier studies,[32] and we chose to calculate with a lower risk reduction value of 29%. This leads to an absolute risk reduction of DVT of 8%, which is higher than the suggested minimum clinically relevant absolute risk reduction of 4%.[9 32] The sample size calculations for the study are based on the following estimates and parameters: alpha=0.05, 1-beta=90%, estimated incidence of DVT=28%, the smallest relative risk reduction to detect=29%. The sample size needed using these study parameters is 597 in each group. Expecting a drop-out rate of 5%–10% and to compensate for the uncertainty of the final case-mix distribution, we aim to include 1400 patients in our study, around 1000 AF patients and 400 ATR patients.

## Statistical analysis plan
### Descriptive statistics
The flow of patients through the trial will be displayed according to the Consolidated Standards of Reporting Trials statement. The number of patients screened; number of ineligible patients and the reasons why; number of eligible patients not providing consent and the number of eligible patients subsequently randomised will be presented. The characteristics of the screened population, the ineligible participants and eligible participants who consent and do not consent will be summarised. Information regarding the number of patients who are surgically treated versus those who are treated non-operatively will be presented. Data on patient eligibility and reasons for withdrawal from treatment or the trial (when stated) will be summarised. Baseline patient characteristics will be summarised using descriptive statistics; counts for categorical variables and mean/median and inter-quartile range for continuous variables.

### General data handling
All analyses will be conducted blinded for the treatment allocation and with pseudonymised data. All statistical tests will be two-sided with an α of 0.05. Primary analyses will be by intention-to-treat. However, per-protocol and as-treated analyses will also be performed. The final pseudonymised data set will be stored separate from the pseudonymisation key on secure hospital and/or university servers, in order to preserve patient confidentiality. Forms completed by the patients are delivered at the end of the trial to the central study organisation.

### Data analyses
#### Primary outcome
Equal distribution will be investigated among the two treatment groups for the factors weight-bearing, anticoagulant treatment, time to treatment, surgery versus no surgery and patient characteristics. The primary outcome will be presented as ORs and 95% CI. For skewed data, both unadjusted and adjusted results will be presented using logistic regression for the factors weight-bearing, anticoagulant treatment, time to treatment, surgery versus no surgery and patient characteristics. Absolute reductions in risk of VTE will be calculated from these values, as presented in the reference with similar study design.[32]

#### Secondary outcome
Group differences for ordinal secondary data, such as the patient-reported data, will be analysed by non-parametric tests. Group differences for index scale, ratio scale and continuous secondary outcome will be analysed with appropriate parametric or non-parametric tests. Subgroup analyses will be performed to determine whether certain types of patients might gain greater or less benefit from IPC. The effect of treatment allocation on the primary outcome subdivided by key baseline variables and adjusted for the factors (weight-bearing, anticoagulant treatment, time to treatment, surgery vs no surgery, compliance and age) will be calculated. Subgroup analyses will be performed by observing the change in log-likelihood when the interaction between the treatment and the subgroup is added into a logistic regression model.

### Missing data
Missing data will be investigated and analysed. Reasons for missing data will be stated and summarised where possible. Sensitivity analyses will be performed in order to test the strength and consistency of the final results.

### Monitoring/auditing
Research nurses educated in monitoring research studies will perform study monitoring during the trial. Auditing of the trial will be performed by Karolinska Trial Alliance, which is an independent organisation from the investigators and sponsor.

## Safety
### Risk and side effect
All included patients have an increased risk of developing DVT and PE. There will be no general recommendation for included patients for administrating pharmacoprophylaxis, in line with international guidelines.[33 34] The steering committee advice only to prescribe VTE-prophylactic drugs to patients with other identifiable risk factors such as previous DVT/PE and usage of birth-control pills. The use of pharmacoprophylaxis will, however, not be an exclusion criterion. Patients with prescription of anticoagulants will be adjusted for in the statistical analysis.

### Adverse events and complications
Adverse events in connection with the usage of the intervention (IPC and orthosis) or casting will be noted when reported in the study protocol and referred to the responsible physician if needed. Complications such as infection, fracture dislocation, tendon rerupture, reoperation or serious life-threatening adverse events will be captured within the clinical injury treatment protocol of the local hospital. Minor or major complications may also be noted through the detailed study protocol and referred to the responsible physician if needed.

### Patient and public involvement
A patient representative is on the steering committee of the research project to help in planning and improving the relevance, quality and validity of the research. The patient representative will also receive appropriate training for the job and will actively participate in the development of recommendations and guidelines.

## DISCUSSION
### Overall aim of the trial
There is an urgent need for more efficient and safer VTE prophylactics, especially for lower limb immobilised

patients, according to major health service authorities.[33 34] The overall aim of STOP leg clots is to investigate whether or not adjuvant IPC treatment is reliably effective in outpatient lower leg immobilised patient. We therefore chose two types of injuries, ATR and AF, which constitute a large portion of patients with lower leg immobilisation that also have a significantly increased risk of VTE.[35] Moreover, patients with ATR and AF have demonstrated low-effectiveness or non-effectiveness of LMHW prophylaxis also in relation to the risk of bleeding complications.[13 14 35] In the perspective that different hospitals have different clinical traditions and indications for surgery of ATR and AF, we have chosen to include both operated and non-operated ATR/AF, since both groups are equally served by VTE prophylaxis. Earlier studies have shown a just as high incidence of VTE in non-operated as in operated ATR/AF,[36] further indicating a benefit of demonstrating a VTE risk reduction in both operated/non-operated lower limb immobilised patients.

## Pharmacological prophylaxis

In the initial planning of the trial, the intervention was thought to be compared with patients without any pharmacological prophylaxis in order to study the true VTE preventive effects of the IPC device. However, there were some sites that did not feel comfortable with this comparator group despite the lack of strong VTE-preventive evidence of pharmacological prophylaxis. In order to ensure a higher inclusion rate into the trial, we therefore included the available data for antithrombotic drugs in our power analysis. The primary treating physician will decide if and any prophylactic antithrombotic drugs will be administrated to the patient during the immobilisation period, which will be decided prior to randomisation. LMWH is the most common drug prescribed for usually 10–14 days, whereas direct oral anticoagulant drugs are seldomly used as pharmacological prophylaxis in previously healthy patients. The trial advices against antithrombotic prophylactic drugs if there are no known risk factors for thrombosis and have provided each participating hospital with the international guidelines of whom and when pharmacological treatment should be administered.

## Compliance

Compliance to the allocated treatment, specifically number of days and hours of IPC usage, is key in understanding if and how the intervention provide an effect on the primary outcome. This will be monitored by patient self-assessment diaries together with a built-in cumulative timer in the IPC device. After each week, patients will receive an email and be prompted to transfer the data into the REDCap system. American College of Chest Physicians recommend that efforts should be made to achieve 18 hours of daily compliance with IPC in hospitalised patients.[37] However, there are no recommendations for mobile outpatients. In this study, we will recommend at least 10 hours of bilateral IPC treatment per 24 hours.

However, patients should not be more sedentary than otherwise in order to comply with treatment. Patients will be considered compliant with the intervention if used for at least 30 hours/week throughout the duration of lower limb immobilisation.

## Endpoints

Compression Doppler ultrasonography is the investigation of choice to confirm the diagnosis of DVT since it is non-invasive, acceptable to patients, easily repeated several times and uses equipment widely available.[38] Thus consent, accrual and adherence in a trial would more likely be achieved than with alternatives such as phlebography. CDU screening is performed at the end of immobilisation, that is, after 6–8 weeks. CDU screening will detect both symptomatic as well as asymptomatic DVTs. However, it will be difficult to distinguish between the symptomatic and asymptomatic DVTs since calf pain related to DVT can be confounded by the lingering pain from the AF or ATR.

## Clinical recommendations

Most DVTs in this trial will be asymptomatic DVT', only demonstrated by the CDU screening, which will serve as a surrogate outcome for the total VTE burden. The natural history of DVTs seems to be, in the majority of cases, the development of a thrombus in the distal veins of the calf that can extend proximally. However, recent studies demonstrate that even asymptomatic DVTs are associated with reduced patient-reported outcomes 1 year after initial injury[39] and can extend proximally and cause PE.[40] This trial will provide further insight on whether exhibiting a DVT affects patient-reported outcome, as this is one of the secondary outcomes, as well as a cost–benefit analysis of the intervention. The data will also be analysed to give results on which patients are most likely to exhibit a benefit from the mechanical IPC prophylaxis. If the usage of IPC devices demonstrates a significant reduction of DVTs, we will recommend IPC devices for prophylactic treatment during lower limb immobilisation.

## Ethics and dissemination of results

Ethical approval was obtained by the ERB in Stockholm, Sweden, Dnr 2016/1573-31. The results of this study will provide evidence-based treatment protocol for leg immobilised patients. The study results will be submitted for publication to international, peer-reviewed journals, regardless of whether the results are positive, negative or inconclusive in relation to the study hypothesis.

### Author affiliations
[1]Department of Orthopedic Surgery, Karolinska University Hospital, Stockholm, Sweden
[2]Department of Molecular Medicine and Surgery, Karolinska Institute, Stockholm, Sweden
[3]Department of Orthopaedics, Eksjö, Region Jönköping County and Department of Biomedical and Clinical Sciences, Linköping University, Linköping, Sweden
[4]Division of Orthopaedics and Biotechnology, Karolinska Institute Department of Clinical Sciences Intervention and Technology, Huddinge, Sweden

[5]Department of Clinical Sciences, Danderyd Hospital, Karolinska Institute, Stockholm, Sweden
[6]Department of Surgical Sciences, Uppsala University Hospital, Uppsala, Sweden
[7]Department of Orhopaedics, Sahlgrenska University Hospital, Göteborg, Sweden

**Contributors** All authors have contributed to the design of the present trial protocol. PWA is the primary investigator and initiated the study. SS, BA, HEB, ED-A, KJ and KNH are all helping with study implementation. The protocol was drafted by PWA, SS and refined by BA, HEB, ED-A, KJ and KNH. SS was responsible for drafting of the manuscript. All authors have read and approved the final manuscript.The STOP leg clots trial is currently including patients from the following centres (local study personnel in parenthesis): Akademiska Universitetssjukhuset (Kenneth Jonsson, Catharina Strömstedt), Danderyd Hospital (Erica Domeij-Arverud, Sophia Grindberg, Paula-Therese Kelly, Marie Ax), Gävle Sjukhus (Irina Kolioumpakina Akrioti, Hans Peter Bögl), Helsingborgs lasarett (Anna-Karin Svensson), Höglandssjukhuset Eksjö (Björn Alkner), Karolinska Universitetssjukhuset (Paul Ackermann, Susanna Aufwerber, Luigi Belcastro, Hans Berg, Johanna Flodin, Malin Heijkenskjöld, Simon Svedman, Annukka Saarensilta), Norrtälje Sjukhus (Per-Anton Svensson), Sahlgrenska Universitetssjukhuset (Katarina Nilsson-Helander, Niklas Nilsson), Södersjukhuset (Lasse Lapidus, Elisabeth-Skogman, Annika Tiliander), Uddevalla Sjukhus/Norra Älvsborgs Sjukhus (Anna Ludvigsson, Tina Zorko), Östersunds sjukhus (Harry Mitchell, Alexander De Val Olsson, Jörgen Larsson).

**Funding** The study is financed by the Swedish research council (Dnr: 2017-00202). https://www.vr.se/english/about-us/contact.html.

**Disclaimer** DJO Global is the manufacturer of the IPC devices used in the study and have made them available for the study sites throughout the study. Disposable materials (cuffs and connective tubing) are bought from the DJO Global. DJO Global has no influence in the conduct of the trial or publication of the results.

**Competing interests** None declared.

**Patient and public involvement** Patients and/or the public were involved in the design, or conduct, or reporting, or dissemination plans of this research. Refer to the Methods section for further details.

**Patient consent for publication** Not required.

**Provenance and peer review** Not commissioned; externally peer reviewed.

**ORCID iD**
Simon Svedman http://orcid.org/0000-0002-8872-7209

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
