## [Reviewer comments · BMJ Open]

ARTICLE DETAILS

TITLE (PROVISIONAL)	STOP Leg Clots - Swedish multicenter Trial of Outpatient Prevention of Leg Clots: study protocol for a randomized controlled trial on the efficacy of intermittent pneumatic compression on venous thromboembolism in lower leg immobilized patients
AUTHORS	Svedman, Simon; Alkner, Björn; Berg, Hans; Domeij-Arverud, Erica; Jonsson, Kenneth; Nilsson Helander, Katarina; Ackermann, Paul

VERSION 1 – REVIEW

REVIEWER	Davies, AH Imperial College London, Vascular
REVIEW RETURNED	28-Oct-2020

GENERAL COMMENTS	1. Think need to be sure not over estimated DVT risk2. To maximise pick up need to clarify timing of duplex examination3. Think outcome could be clarified to be clear asymptomatic and symptomatic DVT and symptomatic PE4. State compression ultrasound, sometimes this means just point compression, this needs to be clarified5. Standard care in Sweden needs clarity, is it just LMWN ? Any DOACs
---

REVIEWER	Cannegieter, Suzanne Leiden University Medical Centre, Leiden, Department of Thrombosis and Hemostasis
REVIEW RETURNED	29-Nov-2020

GENERAL COMMENTS	This manuscript describes the protocol of a multicentre randomised controlled superiority trial conducted across Sweden. Eligible patients (700 patients/arm) with an acute ankle fracture or Achilles tendon rupture will be randomized to either intermittent pneumatic compression (IPC) during the immobilization period or treatment-as-usual. The primary outcome will be the total VTE-incidence during the leg immobilization period, approximately 6-8 weeks, as assessed by screening compression duplex ultrasound at removal of leg immobilization and/or clinically diagnosed pulmonary embolism. The trial addresses an important clinical question, as VTE incidence is still high in these patients, while a good way to prevent it is still lacking. Furthermore, there may be additional benefits related to the IPC treatment. The protocol is quite
---

	straightforward, nevertheless I have some comments and questions: It is not completely clear when the randomization will take place. If not shortly after the injury, how will these patients be treated before randomization, and what will happen to patients who develop a VTE before randomization takes place? Exposure to IPC will be blinded for the people who perform the ultrasound but how will the blinding be guaranteed? Surely it will be difficult to keep patients from talking about their experiences? A problem that the investigators also raise is the fact that asymptomatic events will be the main outcome event. Their relevance, as they acknowledge, is not clear, so can the authors indicate how the results will be interpreted? What will be the clinical advice in case the IPC shows a significant benefit? The follow-up ends at the moment the immobilization is completed, when the ultrasound is performed. However, several studies have shown that VTE not seldomly occurs AFTER the immobilisation period, which generally coincides with the end of prophylactic measures. Is there a way to extend follow-up and take this important period also into account? The investigators plan to provide odds ratios, adjusted for the factors weight-bearing, anticoagulant treatment, time to treatment, surgery vs. no surgery and patient characteristics. I cannot completely follow this reasoning: first, why is this necessary when the treatment is randomized? Secondly, some of these factors are in the causal pathway: can it be that doctors will provide patients who were randomized to standard treatment with anticoagulation? Or perhaps IPC will influence the amount and speed of weight-bearing? Can the authors comment on this possibility and how it will influence the results? If patients are not compliant with the treatment, it is unlikely that they will keep the diaries. Why are the diaries necessary in the first place when compliance is measured automatically? Will compliance also be taken into account as expressed in hours, rather than in a dichotomous way using 30 hours as a cut-off? Minor comment Ref 7 should probably be replaced by the one from these authors that was published in the NEJM, which describes the effect of LMWH
--	---

REVIEWER	Hunt, Beverley Guy's & St Thomas' Foundation Trust, Dept of Haematology
REVIEW RETURNED	04-Jan-2021

GENERAL COMMENTS	This is a study using intermittent pneumatic compression in the recovery of those with immobilised lower limbs with a primary endpoint of looking at rates of venous thromboembolism. This is a fascinating area and I applaud the authors for undertaking this work. I have a few concerns about this paper and addressing them I think will improve the quality of the messaging. as a general comment there is inadequate detail on the various parameters
---

	being studied for a reader who is not an orthopaedic surgeon to understand the study. 1) The primary outcome needs to include deaths due to VTE as well as clinical events. I very much hope there wont be any but it is a normal inclusion in a total VTE endpoint. 2) the secondary endpoint term "profibrinolytic substances" would be better replaced through the document with the term "fibrinolysis" Introduction. The content seems very simplistic, and lacks the necessary details to give the reader new to this area the adequate background information and uses old references e.g. stating VTE is a "dangerous condition" needs applification of what is meant here - with figures of morbidity and mortality. There is under referencing and many of the references are very old and out of date e.g. reference 10 refers to NICE guidance in 2010, which has been updated in 2018. e.g Sentence 2 needs a reference e.g last para p4 -need more details of recently published studies such as number of patients and statistical significance of the results of the various studies. Method Please explain how IPC will be applied to the patients affected leg if they are wearing a plaster? Will those who are wearing removable boots take them off during the time of IPC? Another point that is not mentioed until p 10 ius that it appears some of the patients will receive pharmacological thromboprophylaxis. As there is now evidence that this has a positive effect, then surely the use of this should be standardised in this study, otherwise this will be a confounding factor? Do the authors have enough patients included to look at the effects of combined LMWH with IPC vs IPC alone vs LMWH alone? There are many secondary end points but no justification is given either in the introduction or methods for their inclusion. Why is post-injury weight-bearing included? What is microdialysis of Achilles tendon healing? How does the heel -rise endurance test add to the study? More details need to be given about the markers of coagulation and fibrinolysis. How will the samples be collected? will they be processed within a set time (fibrinolytic factors can deteriorate over time). Blood flow quantification. Will this be done on and off IPC?
--	--

VERSION 1 – AUTHOR RESPONSE

Reviewer: 1

Dr. AH Davies, Imperial College London

Comments to the Author:

1. Think need to be sure not over estimated DVT risk

- We agree with the reviewer, but we already have preliminary data on the high incidence of ultrasound Doppler verified DVTs in leg immobilized patients with Achilles tendon ruptures and with ankle fractures.

2. To maximise pick up need to clarify timing of duplex examination

- We thank the reviewer and we have considered this. As long as the patient is immobilized in a cast or orthosis, the risk of VTE is imminent. Compression ultrasound examination is therefore performed at removal of leg immobilization. Since the time of leg immobilization vary among the patients, the timing of the duplex investigation will therefore also vary.

3. Think outcome could be clarified to be clear asymptomatic and symptomatic DVT and symptomatic PE

- We thank the reviewer for the suggestion and have in the revised manuscript clarified the primary outcome as suggested.

4. State compression ultrasound, sometimes this means just point compression, this needs to be clarified

- As suggested by the reviewer we have in the revised manuscript added the information that the compression ultrasound includes a whole leg investigation. There is also details on which veins that will be investigated in the manuscript.

5. Standard care in Sweden needs clarity, is it just LMWH ? Any DOACs

- In line with the comment of the reviewer we have in the revised manuscript clarified the standard of care treatment including LMWH and DOACs. Since there are different treatment regimens for leg immobilized patients at each individual hospital we will prospectively note the DVT-preventive drug treatments prescribed to the patients and will analyse the effect.

Reviewer: 2

Dr. Suzanne Cannegieter, Leiden University Medical Centre, Leiden, Leiden University Medical Centre

Comments to the Author:

This manuscript describes the protocol of a multicentre randomised controlled superiority trial conducted across Sweden. Eligible patients (700 patients/arm) with an acute ankle fracture or Achilles tendon rupture will be randomized to either intermittent pneumatic compression (IPC) during the immobilization period or treatment-as-usual. The primary outcome will be the total VTE-incidence during the leg immobilization period, approximately 6-8 weeks, as assessed by screening compression duplex ultrasound at removal of leg immobilization and/or clinically diagnosed pulmonary embolism.

The trial addresses an important clinical question, as VTE incidence is still high in these patients, while a good way to prevent it is still lacking. Furthermore, there may be additional benefits related to the IPC treatment. The protocol is quite straightforward, nevertheless I have some comments and questions:

It is not completely clear when the randomization will take place. If not shortly after the injury, how will these patients be treated before randomization, and what will happen to patients who develop a VTE before randomization takes place?

- We thank the reviewer for addressing the question of randomization, which we have discussed in more detail in the revised manuscript. Randomization will occur as soon as possible after the injury. However, since patients may not seek medical care immediately after the injury and also present during weekends or holidays, we decided that screening and randomization of patients may occur up to 10 days after the injury. The number of days from injury to randomization will be noted as well as the treatment. Unfortunately, we are not able to perform an inclusion screening for VTE due to logistic and economic reasons.

Exposure to IPC will be blinded for the people who perform the ultrasound but how will the blinding be guaranteed? Surely it will be difficult to keep patients from talking about their experiences?

- We agree that the discussion about blinding can be expanded on and have done so in the revised manuscript. The study nurse will remove the lower-leg immobilization of the patient before the ultrasound examination and inform the patient not to disclose the treatment for the sonographer. Moreover, the sonographer is informed only to perform the examination and not to discuss the treatment with the patient.

A problem that the investigators also raise is the fact that asymptomatic events will be the main outcome event. Their relevance, as they acknowledge, is not clear, so can the authors indicate how

the results will be interpreted? What will be the clinical advice in case the IPC shows a significant benefit?

- We thank the reviewer for the suggestion and have in the revised manuscript elaborated on the clinical advice that can be made. Since many clinicians prescribe LMWH, which have low or no efficacy to prevent DVT in lower-limb immobilized patients, an alternative with mechanical prophylaxis, if clinically efficient, would be a welcome treatment alternative without adverse events of bleeding.

- Most DVTs diagnosed in this study will be asymptomatic DVTs demonstrated by compression Duplex ultrasound (CDU) screening, which will serve as a surrogate outcome for the total VTE burden. The natural history of DVT seems to be, in the majority of cases, the development of a thrombus in the distal veins of the calf that can extend proximally—the so-called ascending pattern of thrombus extension. Moreover, recent data show that also asymptomatic DVTs to a high degree can cause thrombus extension and be lethal. Therefore, we will use the CDU diagnose as a surrogate of VTE.

- This study will provide further insight on how any DVT affect patient-reported outcome, which is a secondary outcome measure. Another secondary outcome is a cost-benefit analysis, which will be performed within this trial. Data from the trial will also be used to assess which patients are most likely to exhibit a benefit from mechanical prophylaxis.

- Therefore, in case the IPC shows a significant reduction of DVTs we will recommend mechanical prophylaxis to lower-limb immobilized patients. Data from the trial will demonstrate if any subgroups are more likely to benefit. The cost-benefit analysis will assess at what price the mechanical prophylaxis can benefit a larger group of lower-limb immobilized patients. The follow-up ends at the moment the immobilization is completed, when the ultrasound is performed. However, several studies have shown that VTE not seldomly occurs AFTER the immobilisation period, which generally coincides with the end of prophylactic measures. Is there a way to extend follow-up and take this important period also into account?

- We agree with the reviewer that this is an important research subject. The vast majority of trial data pertaining to leg immobilized patients, however, have continued prophylactic anticoagulation for the duration of immobilization in plaster. Therefore, in this study we choose as suggested by international reviewers to use CDU at the time of removal of leg immobilization. Our budget restricted repeated CDU investigations for this trial. However, we will consider in a separate trial to investigate repeated CDU after removal of leg immobilization.

The investigators plan to provide odds ratios, adjusted for the factors weight-bearing, anticoagulant treatment, time to treatment, surgery vs. no surgery and patient characteristics. I cannot completely follow this reasoning: first, why is this necessary when the treatment is randomized?

- We agree with the reviewer and have in the revised manuscript expanded on the statistical analysis plan. Treatment with mechanical prophylaxis, IPC, is randomized. We will check for equal distribution of the factors among the two treatment groups for the factors weight-bearing, anticoagulant treatment, time to treatment, surgery vs. no surgery and patient characteristics. We plan to provide odds ratios and 95% confidence intervals, for both unadjusted and adjusted using logistic regression for the factors (factors weight-bearing, anticoagulant treatment, time to treatment, surgery vs. no surgery and patient characteristics), with skewed distribution among the treatment groups. Secondly, some of these factors are in the causal pathway: can it be that doctors will provide patients who were randomized to standard treatment with anticoagulation? Or perhaps IPC will influence the amount and speed of weight-bearing? Can the authors comment on this possibility and how it will influence the results?

- We thank the reviewer for these insightful comments, which have been expanded on in the revised manuscript. We have provided each hospital with the international guidelines of whom and when pharmacological recommendations should be administered. However, we also do not interfere with the decision of the physician treating the patient and do not remove or add e.g. LMWH. Therefore, as mentioned above we will control for the equal distribution of the factors (weight-bearing, anticoagulant treatment, time to treatment, surgery vs. no surgery and patient

characteristics) among the two treatment groups and will provide both unadjusted and adjusted data for skewed variables.

- Moreover, subgroup analyses will be performed to determine whether certain types of patients might gain greater or less benefit from IPC. We will estimate the effect of treatment allocation on the primary outcome subdivided by key baseline variables and adjusted for the factors (weight-bearing, anticoagulant treatment, time to treatment, surgery vs. no surgery, compliance and age). Subgroup analyses will be performed by observing the change in log-likelihood when the interaction between the treatment and the subgroup is added into a logistic regression model. We will determine whether there is significant heterogeneity between these subgroups.

If patients are not compliant with the treatment, it is unlikely that they will keep the diaries. Why are the diaries necessary in the first place when compliance is measured automatically? Will compliance also be taken into account as expressed in hours, rather than in a dichotomous way using 30 hours as a cut-off?

- In line with the comment of the reviewer we have in the revised manuscript clarified the use of diaries. The diaries give more information as they give day-by-day information while the pump only records cumulative time used. The diaries furthermore function as a physical backup, if for any reason the timer is reset by mistake while the patient uses the pump at home. A prior study has demonstrated a high correlation between pump usage from diaries compared with pump recorded time.

Minor comment

Ref 7 should probably be replaced by the one from these authors that was published in the NEJM, which describes the effect of LMWH

- We thank the reviewer for noticing this and have adjusted accordingly.

Reviewer: 3

Dr. Beverley Hunt, Guy's & St Thomas' Foundation Trust

Comments to the Author:

This is a study using intermittent pneumatic compression in the recovery of those with immobilised lower limbs with a primary endpoint of looking at rates of venous thromboembolism. This is a fascinating area and I applaud the authors for undertaking this work.

I have a few concerns about this paper and addressing them I think will improve the quality of the messaging. as a general comment there is inadequate detail on the various parameters being studied for a reader who is not an orthopaedic surgeon to understand the study.

1) The primary outcome needs to include deaths due to VTE as well as clinical events. I very much hope there wont be any but it is a normal inclusion in a total VTE endpoint.

- We thank the reviewer for the comment. Number of deaths has not been planned as a primary outcome due to low power but will however be presented. This information has been added to the revised manuscript.

2) the secondary endpoint term "profibrinolytic substances" would be better replaced through the document with the term "fibrinolysis"

- We appreciate the input of the reviewer and have made changes accordingly.

Introduction. The content seems very simplistic, and lacks the necessary details to give the reader new to this area the adequate background information and uses old references e.g. stating VTE is a "dangerous condition" needs application of what is meant here - with figures of morbidity and mortality. There is under referencing and many of the references are very old and out of date e.g. reference 10 refers to NICE guidance in 2010, which has been updated in 2018. e.g Sentence 2 needs a reference e.g last para p4 -need more details of recently published studies such as number of patients and statistical significance of the results of the various studies.

- We thank the reviewer for the important feedback and have made extensive revisions accordingly. Method Please explain how IPC will be applied to the patients affected leg if they are wearing a plaster? Will those who are wearing removable boots take them off during the time of IPC?
- We agree with the reviewer that the application of the intervention requires a more detailed description and have in the revised manuscript extended the information. Patients randomized to IPC will wear removable boots, which fits with the IPC as shown in an earlier study [1]. IPC does not seem to work well under plaster cast [2]. Patients wearing removable boots have the opportunity to take them off during one hour daily, and this does not affect the rate of DVT as demonstrated earlier [1].
- 1. Arverud E, Labruto F, Latifi A, Nilsson G, Edman G, Ackermann P.W Intermittent pneumatic compression reduces the risk of deep vein thrombosis during post-operative lower limb immobilisation. A prospective randomised trial of acute Achilles tendon ruptures. Bone Joint J. 2015 May;97-B(5):675-80. doi: 10.1302/0301-620X.97B5.34581
- 2. Arverud E, Greve K, Bring D, Nilsson G, Ackermann P.W Can Foot Compression under Plaster Cast prevent deep Vein Thrombosis during lower limb immobilization? Bone Joint J. 2013 Sep;95-B(9):1227-31. doi: 10.1302/0301-620X.95B9.31162.

Another point that is not mentioned until p 10 is that it appears some of the patients will receive pharmacological thromboprophylaxis. As there is now evidence that this has a positive effect, then surely the use of this should be standardised in this study, otherwise this will be a confounding factor? Do the authors have enough patients included to look at the effects of combined LMWH with IPC vs IPC alone vs LMWH alone?

- We appreciate the reviewers comment and have highlighted the use of LMWH already earlier in the manuscript and expanded on the use of LMWH in the revised manuscript. There is still no evidence in the scientific literature that LMWH provides a significantly strong protection against DVT or PE during lower leg immobilization. However, many hospitals in Sweden have their own medical memorandum for when to use LMWH prophylaxis and did not want to take part in the trial if we did not allow for the individual physician to choose administration or not of LMWH.
 - The poweranalysis has therefore been adjusted for the preventive effects of LMWH, which are presented in the scientific literature. This study has, however, not been powered to look at the effects of combined LMWH with IPC vs IPC alone vs LMWH alone.
 - Moreover, as commented to reviewer 2 the primary outcome will be presented both unadjusted and adjusted for LMWH treatment.
- There are many secondary end points but no justification is given either in the introduction or methods for their inclusion. Why is post-injury weight-bearing included? What is microdialysis of Achilles tendon healing? How does the heel -rise endurance test add to the study? More details need to be given about the markers of coagulation and fibrinolysis. How will the samples be collected? will they be processed within a set time (fibrinolytic factors can deteriorate over time).
- These are excellent points raised by the reviewer, which we have expanded on in the revised manuscript.
- Blood flow quantification. Will this be done on and off IPC?
- Yes, that is correct. Quantification of blood flow with Doppler ultrasound will be performed without and with IPC to demonstrate the haemodynamic features of the intervention.

VERSION 2 – REVIEW

REVIEWER	Cannegieter, Suzanne Leiden University Medical Centre, Leiden, Department of Thrombosis and Hemostasis
REVIEW RETURNED	18-Mar-2021
GENERAL COMMENTS	Many thanks for the clear replies on my earlier comments. I still have a few more small questions:

	With regard to the randomization: I'm still not completely clear on this: Am I right that it can happen that patients are randomized to the treatment arm but have to wait a couple of days to receive it? What happens when they develop a VTE in this time window? I understand that the investigators plan to advice on IPC when a significant effect is established. What needs to be taken into account for this advice, as well as for the cost-effectiveness analysis is that such an effect is likely not significant for symptomatic thrombosis, since this incidence can be a factor 10 smaller (this is the problem of asymptomatic events), i.e., when this is applied in clinical practice. As a suggestion to extend the follow-up a little bit: would it be possible to telephone patients to ask for symptomatic events, for example 3 months after inclusion? I am still a bit worried on the anticoagulant treatment: if this is provided to many patients in the comparison arm and to hardly anyone in the intervention arm, it will be quite difficult to separate the effects. Furthermore, I would suggest not to adjust for factors in the causal pathway (mediators), as this may lead to reduction of the effect associated with the treatment, unless this is done to explore the effect of these factors on the outcomes.
--	---

VERSION 2 – AUTHOR RESPONSE

Reviewer: 2

Dr. Suzanne Cannegieter, Leiden University Medical Centre, Leiden, Leiden University Medical Centre

Comments to the Author:

Many thanks for the clear replies on my earlier comments. I still have a few more small questions:

1. With regard to the randomization: I'm still not completely clear on this: Am I right that it can happen that patients are randomized to the treatment arm but have to wait a couple of days to receive it? What happens when they develop a VTE in this time window?

- This is not correct. When patients are randomized they will receive the intervention on the same day. However, we decided that the inclusion into the study may occur within 10 days after injury. This is due to logistic reasons concerning that patients do not present on the day of injury and that patients cannot be screened during weekends and holidays. Thus, the choice of inclusion within 10 days is a compromise between the risk of VTE and the effort to maximise patient inclusion. For example, a patient with an ankle fracture presents on a Saturday to the emergency with a swollen ankle. If the attending doctor decides that the ankle fracture is a case for surgery the patient may be sent home to reduce swelling before surgery, which is planned for later during next week. On Monday the patient is screened by research staff and later during the week, eg. Thursday the patient is operated on. Post-surgery, the patient will be randomized and receive the intervention.

- If a patient develops a VTE before randomization the patient will be excluded from the study.
- The number of days before the start of the intervention will be noted and will be analysed separately if this will affect the intervention.

2. I understand that the investigators plan to advice on IPC when a significant effect is established. What needs to be taken into account for this advice, as well as for the cost-effectiveness analysis is that such an effect is likely not significant for symptomatic thrombosis, since this incidence can be a factor 10 smaller (this is the problem of asymptomatic events), i.e., when this is applied in clinical practice.

- We agree with the reviewer that we cannot be 100% certain, if a significant effect of IPC is demonstrated, that this will apply also to symptomatic thrombosis. The primary outcome, compression Duplex ultrasound (CDU) screening detected DVTs, will be mostly asymptomatic, but will serve as a surrogate outcome for the total VTE burden. Since the natural history of DVT, in the majority of cases, includes a development of a thrombus in the distal veins of the calf that can extend proximally, the CDU detected DVTs will presumably reflect the total VTE burden.
- The effect of IPC on CDU detected DVTs will be supplemented by potential effects on patient-reported outcomes, which may serve as a surrogate variable for potential symptoms, see below.

As a suggestion to extend the follow-up a little bit: would it be possible to telephone patients to ask for symptomatic events, for example 3 months after inclusion?

- We thank the reviewer for the suggestion of a longer follow-up. At present we follow-up the patients at 6 and 12 months with validated patient-reported outcomes (eg. ATRS). We have already previously demonstrated that patients with non-symptomatic DVTs (CDUetected) report worse outcome, compared to patients with no DVT, as analysed with ATRS at 12 months (Svedman S. et al.). Based on the suggestion from the reviewer we plan also to include questions about symptomatic events; if the patient has been subjected to an VTE during the follow-up period, if this is the same VTE that has previously been reported/discovered during the initial follow-up or if this VTE was found after the initial follow-up. This information has been added to the manuscript. We hope that this might give additional information about VTE that was not discovered at the initial follow-up.

- Reference: Svedman S, Edman G, Ackermann PW Deep venous thrombosis after Achilles tendon rupture is associated with poor patient-reported outcome. *Knee Surg Sports Traumatol Arthrosc.* 2020 Oct;28(10):3309-3317. doi: 10.1007/s00167-020-05945-2. Epub 2020 Apr 20.)

3. I am still a bit worried on the anticoagulant treatment: if this is provided to many patients in the comparison arm and to hardly anyone in the intervention arm, it will be quite difficult to separate the effects. Furthermore, I would suggest not to adjust for factors in the causal pathway (mediators), as this may lead to reduction of the effect associated with the treatment, unless this is done to explore the effect of these factors on the outcomes.

- We greatly thank the reviewer for the concern and efforts to improve our study. With support from the available literature, we believe that anticoagulant prophylaxis will not play a crucial role in VTE prevention and that we have sufficient power to demonstrate differences between the study arms if the intervention has the proposed effect.
- We have, due to a concern of the funding agency, looked at the incidence of CDU detected DVTs in the non-intervention arm with and without pharmacological prophylaxis. We found that patients who received pharmacological anticoagulant treatment (n=57) exhibited a DVT incidence of 45.6%, compared to patients without pharmacological prophylaxis (n=46), who demonstrated a DVT incidence of 47.8%. This is of course a very small number of patients, but it is in line with previous research demonstrating weak, or no effect, of pharmacological DVT-prophylaxis for ankle fracture and Achilles tendon rupture patients.
- Regarding the question of adjusting for a skewed distribution of anticoagulant treatment between the two groups we do not believe that this will be a problem since we have a great number of patients this will render a non-significant difference of the percentage of anticoagulant treated patients in each group. Moreover, the decision to initiate antithrombotic treatment will be done before the randomization of the patient. In our data analyses, however, we will if data are skewed present both unadjusted and adjusted results according to our decisions made together with our statistician.

VERSION 3 – REVIEW

REVIEWER	Cannegieter, Suzanne Leiden University Medical Centre, Leiden, Department of Thrombosis and Hemostasis
REVIEW RETURNED	28-Apr-2021
GENERAL COMMENTS	All my remaining questions have been answered satisfactorily.